# Which Surgical Hand Preparation Method Is More Effective? A Comparative Study of Hand Rub and Hand Scrub

**DOI:** 10.3390/nursrep15070242

**Published:** 2025-07-02

**Authors:** Claudia Olivia, Kusman Ibrahim, Titis Kurniawan

**Affiliations:** 1Nursing Program, Faculty of Nursing, Universitas Padjajaran, Sumedang 45363, West Java, Indonesia; 2Santo Borromeus Hospital, Bandung 40132, West Java, Indonesia; 3Department of Medical-Surgical Nursing, Faculty of Nursing, Universitas Padjajaran, Sumedang 45363, West Java, Indonesia; k.ibrahim@unpad.ac.id

**Keywords:** surgical site infection, surgical hand preparation, hand rub, hand scrub, effectiveness

## Abstract

Surgical Site Infection (SSI) can be a devastating complication, leading to increased morbidity, mortality, and healthcare costs. Pre-surgical hand preparation is an effective strategy to prevent SSI. The two most common pre-surgical hand preparation methods are antimicrobial soap for surgical hand scrub and alcohol-based surgical hand rub. The antimicrobial soap hand scrub remains more commonly used among operating theater staff. However, several studies showed that alcohol-based hand rubs are much more effective than antiseptic soap hand scrubs. **Objective:** This study aimed to compare the effect of the two methods of surgical hand preparation on the number of bacterial colonies. **Methods:** The design of this study was a comparative study with a pre-test and post-test approach in two groups (a surgical hand scrub using 4% chlorhexidine soap group and a surgical hand rub using 70% ethyl alcohol and 2.5% chlorhexidine group). Hand smear sampling was performed before surgical hand preparation (pre-test), immediately after surgical hand preparation (post-test 1), and after the surgery was completed (post-test 2). Seventy-one hand smear samples (35 samples applied the surgical hand rub, and 36 samples applied the surgical hand scrub) were divided into two groups and examined for colony counts in Colony Forming Units (CFU) using the total plate count method. Descriptive and comparative analysis were applied. **Results:** The surgical hand-scrub group had average pre-test, post-test 1, and post-test 2 colony counts of 0.202 CFU/cm^2^, 0.007 CFU/cm^2^, and 0.016 CFU/cm^2^, respectively, while the surgical hand-rub group had average pre-test, post-test 1, and post-test 2 colony counts of 0.163 CFU/cm^2^, 0.001 CFU/cm^2^, and 0.001 CFU/cm^2^ respectively. Statistical analysis using the Friedman test showed that both methods significantly reduced the number of colonies (*p* < 0.01). Based on the Mann–Whitney test, there was no significant difference between the two groups regarding the number of colonies (*p* > 0.05). **Conclusions:** Surgical hand-scrub and hand-rub have similar effectiveness in reducing and maintaining the number of colonies on hands during surgery, both gram-positive and gram-negative bacteria.

## 1. Introduction

Surgical site infection (SSI) is one of the Hospital Acquired Infections (HAIs) that occur due to surgical procedures [1]. Of all reported HAIs, 20% are identified as SSI cases, with an incidence of 2.5–41.9% worldwide [2]. In low- and middle-income countries, SSIs are the most frequent HAIs, affecting one-third of surgical patients [3]. SSIs can cause a lot of harm, including increased cost of care, negative impact on the patient’s mental health, and reduced quality of life [4,5].

Although surgical teams use sterile gloves, damage from small punctures remains common and can increase the risk of surgical site infection (SSI). A study from Nthumba et al. showed that the average incidence of damaged gloves was 18% during surgery and 35% after two hours, of which 80% of this damage goes unnoticed by the surgical team [6,7,8]. Healthcare workers’ hands are one of the pathways of infection transmission, as pathogenic microorganisms can remain on their hands for 2 to 60 min after contamination [9,10]. Prevention is key to reducing SSIs, and surgical hand preparation is one of the measures that can be taken [11].

The purpose of surgical hand preparation is not only to reduce the total number of microorganisms on the hand surface of the surgical team but also to inhibit bacterial growth during the procedure [12,13]. Two of the most popular hand preparation methods exist: the traditional antimicrobial soap and the alternative alcohol-based liquid [14]. Alcohol-based antiseptic products are much more effective than antiseptic soap products [12]. Moreover, previous research supports this, where alcohol-based hand rubs are more efficacious than conventional hand rubs for surgical antisepsis with sustained efficacy [15]. Alcohol has less antiseptic durability with a limited spectrum, and chlorhexidine is an antiseptic with slower antimicrobial activity but a longer residual effect [16,17].

Practically, surgical hand scrubbing requires water, antiseptic soap with a brush or sponge containing antimicrobial soap, and sterile drying wipes. In contrast, surgical hand scrub uses alcohol-based liquids without requiring manual rinsing or drying [7,12,18]. In addition, the surgical hand scrub requires 20.57 L of water to scrub for 3 min, with an additional 6.8574 L required for every additional minute of scrubbing [19]. Another study showed that traditional handwashing methods that use water substantially require water consumption of about 11 L for each hand scrub [20]. The traditional hand scrub method costs about USD 2, while the waterless hand scrub method costs less than USD 1 [21].

Regarding time effectiveness, surgical hand rub requires a shorter duration than surgical hand scrub. Previous studies have shown that the surgical hand scrub implementation time ranges from 3 to 5 min [11,22,23], while surgical hand rub requires 2–3 min [24]. This time, the effectiveness issue was also confirmed by previous studies, which found that the surgical hand scrub method requires an average time of 264 s, while the surgical hand rub only requires an average time of 160 s [20,21].

A study showed that the surgical hand rub method using 61% ethyl alcohol and 1% chlorhexidine was more effective in reducing the number of bacterial colonies on hands when compared to the surgical hand scrub using 4% chlorhexidine soap [15]. Some other literature shows that both methods are effective for surgical hand preparation in reducing hand colony counts [11,22,25,26,27]. In contrast, another study showed that chlorhexidine antiseptic did not reduce the number of bacteria on hands but instead increased the number after 3 h of handwashing [24]. With the different conclusions from the results of previous studies, this study aimed to compare the effect of the two surgical hand preparation methods in reducing hand colony in the setting of surgical procedures carried out in the operating room with varying operating durations. To our knowledge, this study is the only study that directly compares the two methods in the setting of surgical procedures in the operating theater.

## 2. Materials and Methods

### 2.1. Design and Setting

A comparative study with two groups and repeated measurements was taken before surgical hand preparation (pre-test), immediately after surgical hand preparation (post-test 2), and immediately after the surgery procedures were completed (post-test 3). The study was conducted at the operating theater of a 400-bed capacity private hospital in Bandung, Indonesia, from December 2024 to January 2025. This hospital had ten operating theaters and eight automatic handwashing sink facilities using sterilized reverse osmosis water. Automatic dispensers for alcohol-based antiseptics are available in all operating theaters. The average number of operations in one year at this hospital is 4020 procedures.

### 2.2. Population

The population included in this study consisted of surgical personnel consisting of surgeons, assistant surgical nurses, and scrub (instrument) nurses. This group of personnel is directly involved in the surgery processes in the operating theater and plays an important role in the execution of surgical procedures. In this hospital setting, the assistant surgical nurses are responsible for assisting the surgeon technically during surgery, such as opening surgery access, controlling the bleeding, etc. The scrub nurses are responsible for managing and handling sterilized instruments during surgery.

### 2.3. Sample Size Calculation

The number of samples in this study will be determined using power analysis with the Wilcoxon–Mann–Whitney statistical test (two groups). Previous research conducted by Martin-Villa et al. resulted in an effect size = 0.63, α = 0.05, and a power (1–β) of 0.80 [22]. The calculation resulted in the minimum number of samples required being 68 (34 samples for each group). During the study, there were a total of 75 hand-swab samples. However, two samples failed to meet the surgical hand rub group criteria because one sample was contacted with non-sterile objects before the hand swab was performed at the end of surgery (post-test 2), and the other sample failed in the second sampling process (post-test 1). In the surgical hand scrub group, one sample failed to meet the criteria because of her hand contact with a non-sterile object before the third-hand swab sampling was conducted.

A total of 71 surgical hand preparation procedures were obtained as samples, of which 35 samples were for the surgical hand rub method and 36 samples were for the surgical hand scrub method that met the inclusion criteria, namely hand preparations performed by surgeons, surgical assistant nurses, and scrub (instrument) nurses who performed elective surgeries, clean operations, clean contaminated operations, and surgical hand preparation procedures using the hand rub or hand scrub method. Exclusion criteria included those who used both methods simultaneously, failed swab collection, had open wounds on their hands, or contaminated hands before swab collection.

### 2.4. Data Collection

Before the study, the principal investigator (PI—first author) communicated with the person in charge of the operating theaters. After obtaining participants who met the inclusion criteria, the PI provided research information and asked the participant candidates willing to participate to fill out the informed consent form. The participants in this study were surgical hand preparation procedures performed by surgical operators (surgeons), surgical assistant nurses, and scrub (instrument) nurses.

Staff who performed the surgical hand rub using 70% ethyl alcohol combined with 2.5% chlorhexidine following the recommended procedure were assigned to the surgical hand rub group (R1 group), while staff who performed surgical hand scrub using 4% chlorhexidine following the recommended procedure were assigned to the surgical hand scrub group (R2 group). The participants freely selected either of the two surgical hand preparation methods. To ensure the participants correctly performed the selected hand preparation procedures, the researchers observed participants’ compliance with the surgical hand scrub and surgical hand rub procedures using an observation form. The PI collected the first-hand swab sampling before performing surgical hand preparation (pre-test). The second sample was taken immediately after surgical hand preparation (post-test 1), and the third sample was taken immediately after surgery, and the staff removed sterile gloves (post-test 2) (Figure 1).

The PI performed hand swab sampling using sterile Amies sticks using the zigzag ‘S’ swab method on the hands in the dorsal and ventral areas of the hands, as well as the areas between the fingers and fingertips. The PI then sent the hand swab samples to the laboratory for microbiological examination. The analyst recorded the examination results on the result sheet for data processing (Figure 2). In this study, the microbiological analyst directly inoculated the samples obtained using Amies swabs onto nutrient agar media. After incubation under certain conditions (2–5 days), the growing colonies were counted by measuring the colony growth area on the agar medium surface. The Plate Count Agar (PCA) tools were used to observe the growth and count the number of bacterial colonies using the total plate count method and a calibrated Olympus Brand Microscope. The colony count results were then reported in Colony Forming Units per square centimeter (CFU/cm^2^). If no colonies were observed for 5 days, it was counted as no colonies.

### 2.5. Data Analysis

The descriptive and comparative analysis was applied. The initial descriptive analysis was applied to analyzing the participants’ characteristics, including gender, average age, working experience, profession (surgeon, nurse), and duration of surgery. This analysis was also performed to describe the bacteria colonies in each group.

A normality test was performed on the collected data using the Shapiro–Wilk test before the inferential and comparative analysis. This normality test was chosen because it has high sensitivity, especially in a study with small sample sizes. The Friedman test was performed to see changes in the number of bacterial colonies in each group (surgical hand scrub and hand rub group). Moreover, the Mann–Whitney test was conducted to compare the number of bacterial colonies between the groups.

### 2.6. Ethical Considerations

This study was approved by the health research ethics committee of Santo Borromeus Hospital with number 031/KEPK/XII/2024. All participants in the study received information about the purpose, benefits, and procedures during the study before they decided to participate. The PI also informed the participants that their participation in this study was voluntary and that they could refuse to participate and/or withdraw from the study without any consequences. They were also informed that their identity was kept confidential. All participants who agreed to participate signed a consent form.

## 3. Results

### 3.1. Demographic Characteristics and Surgery

The demographic characteristics of participants in this study include working experience and age, while the characteristics of the operation consist of duration of surgery, profession, and type of surgery.

Table 1 showed that most surgical hand rub group participants were male, while more women were in the surgical hand scrub group. Working experience in both groups was mostly between 11 and 15 years and 16 and 20 years. The average working experience was 13.46 years in the surgical hand rub group and 16.17 years in the surgical hand scrub group. The average age in the surgical hand rub group was 40.91 ± 5.711 years old, and in the surgical hand scrub group, it was 39.19 ± 8.448 years old. The Chi-Square test found that the age distribution of participants from the surgical hand scrub and surgical hand rub groups did not show a significant difference (*p* > 0.05). However, the Mann–Whitney test showed that working experience significantly differed between the two groups (*p* < 0.05).

The duration of surgery in the surgical hand scrub and surgical hand rub groups showed a balanced proportion, where the average duration of surgery in the surgical hand rub group was 105.00 ± 67.475 min, and the surgical hand scrub group was 113.75 ± 63.475 min. The characteristics of the duration of surgery, participant’s profession, and type of surgery in both methods showed the same distribution (*p* > 0.05).

### 3.2. Types of Microorganisms Based on Gram Stain

*Gram-positive* coccus bacteria were the primary microorganism type found in the participants’ hands before both groups underwent surgical hand hygiene (45.45% and 63.41%) (Figure 3). The Friedman test showed a significant difference in colony types at pre-test, post-test 1, and post-test 2 (*p* < 0.01), and the Whitney test showed no significant difference between the two groups regarding colony count according to Gram stain. (*p* value > 0.05) (Table 2).

### 3.3. Bacteria Colony Count in Surgical Hand Scrub and Surgical Hand Rub Groups

In this study, we compared the changes in the number of colonies of both groups before surgical hand preparation (pre-test), immediately after surgical hand preparation (post-test 1), and when the operation was completed (post-test 2). The results are presented in Table 3.

Based on Table 3, in the surgical hand scrub group, the highest colony count was observed in the pre-test (2.89), followed by post-test 2 (1.63), and the smallest in post-test 1 (1.49). Additionally, there was a significant difference in the number of bacteria colonies between the three datasets (*p* < 0.01). Similarly, in the surgical hand rub method, the highest colony count was also observed in the pre-test (2.81), followed by post-test 2 (1.63) and the smallest in post-test 1 (1.56). In addition, there was a significant difference in the number of bacteria colonies between the three datasets (*p* < 0.01).

Further, Post Hoc analysis in both groups showed that the number of colonies in the pre-test was significantly higher than those in post-test 1 and post-test 2. Although the number of colonies in post-test 2 in both groups was higher than those in post-test 1, they were not significantly different (*p* > 0.05).

### 3.4. Comparison of the Difference in Bacterial Colonies of the Surgical Hand Scrub and Surgical Hand Rub Groups

This study also compared the difference (delta) in the number of bacterial colonies between the two groups. The Mann–Whitney test was used to see the difference between the two groups. Test results were presented in Table 4.

The results of the Mann–Whitney test compared the change in the colonies number (delta) from pre-test to post-test 1 (delta 1) and post-test 2 (delta 2), as well as from post-test 1 to post-test 2 (delta 3), between the surgical hand scrub and surgical hand rub groups. The results showed there was no statistically significant difference in the changes in colony number (delta 1, 2, and 3) between both groups (*p* > 0.05).

## 4. Discussion

This study involved participants from various age groups, medical professions, lengths of work experience, and types of surgery. The distribution of variability in characteristics between the two groups did not show significant differences except in terms of work experience, where the surgical hand scrub group tended to have a longer average work experience compared to the surgical hand rub group. This working experience might affect the study results since surgical room personnel with more years of experience may have learned different hand preparation techniques due to changes in practice and guidelines [28]. However, a previous study showed that length of working years and age of personnel did not affect the implementation of surgical hand preparation [29]. The procedure characteristics in this study were the duration of surgery, the profession of the health care worker, and the type of surgery performed. There was no significant difference between the surgical hand scrub and surgical hand rub groups in terms of the duration of surgery, the profession of the healthcare worker, and the type of surgery performed. The duration of surgery is associated with the incidence of recontamination or an increase in the number of bacteria colonies on the hands of personnel. Recolonization is likely to occur before the duration of surgery reaches 5 h [30]. In addition to the similarity between groups in terms of surgery duration, the fact that the surgery duration in our study was less than 5 h may strengthen the idea that surgery duration will not have an impact on the study results.

In the (bacterial) culture we sampled from participant hand swabs prior to hand hygiene practice, we found that the most microorganisms were gram-positive bacteria in the form of cocci. This finding correlates to the fact that staphylococci are the most common commensal bacteria on human skin. Gram-negative bacteria found in 16.90% of participants were likely pathogens due to contact with the environment. The findings in this study align with previous research, which also found a high prevalence of gram-positive bacteria before the use of antiseptics (61.3%) on the hands of surgical room staff and a reduction of 20.26% following the use of antiseptics [31]. Other studies have also shown that most of the pathogens identified on the hands of operating room attendants are gram-positive bacteria [15].

This study showed that both methods were equally effective in reducing and suppressing the number of colonies on the hands of operating room personnel during surgery. The results in this study are in line with previous research, where it was stated that both methods had statistically similar effectiveness (*p* = 0.18) in reducing the number of colonies on the hands of operating room staff, even though the researchers stated that surgical hand rub had a longer effect but was not statistically proven [26]. Other studies also stated that there was no significant difference in bacterial reduction between the two surgical hand preparation methods, but the waterless method was better in terms of humidity [25].

The surgical hand scrub method in this study uses 4% chlorhexidine soap, which will cause the death of bacteria on the hands. Chlorhexidine is the most used antiseptic for surgical handwashing and is a broad-spectrum biocide effective against gram-positive bacteria, gram-negative bacteria, and fungi [32,33]. Previous studies have also shown that 4% chlorhexidine antiseptic effectively reduces and suppresses bacterial growth [34]. Chlorhexidine at low concentrations (0.02–0.06%) can exert bacteriostatic effects by displacing Ca^2+^ and Mg^2+^ ions, resulting in K^+^ leaving the bacterial cell wall. At higher concentrations (>0.1%), chlorhexidine causes leakage of important components inside the cell, resulting in a bactericidal effect through cell lysis and death. In viruses, chlorhexidine has a function to inactivate the virus by changing the permeability of the cell membrane [16].

In this study, the surgical hand rub method was applied using an alcohol-based antiseptic liquid (70% ethyl alcohol combined with 2.5% chlorhexidine). Alcohol refers to the antiseptic and water-soluble chemical compounds ethyl alcohol and isopropyl alcohol, which are highly effective against viruses and bacteria at concentrations of 60–80% [35]. Alcohol works by denaturing proteins, damaging the protective coating of microbes so they do not function. While alcohol can kill bacteria on the skin quickly, it does not have a long-term effect [12,36]. Chlorhexidine is an antiseptic with a long residual effect, and the ability to kill bacteria is slower when compared to alcohol [12].

An increase in colony count was observed during the surgical procedure in both hand preparation method groups. However, the increase in the number of colonies in both groups was not significantly different. Thus, the role of chlorhexidine in suppressing the growth of microorganisms for at least 120 min of surgery is still effective. A previous study also supported this finding and found that the number of bacteria significantly reduced after chlorhexidine-based antiseptic application and after 3 h of use [11].

In the nursing world, hand hygiene is an important effort to prevent HAIs, including in the surgical area [37]. The results of this study provide important implications, especially regarding the implementation of handwashing by nurses before performing surgery. Surgical hand scrub and hand rub have similar effectiveness in reducing the number of colonies on the hands of operating room staff. In both groups, many colonies were found before surgical hand preparation, and the number of colonies was significantly reduced after the surgical hand scrub and hand rub application. This implies that surgical hand preparation methods, either a scrub or a hand rub, are one of the most effective strategies for minimizing SSI. This current study also clarifies the findings from the previous studies debating the effectiveness of the surgical hand rub and hand scrub.

Surgical hand rub can be an effective alternative and equivalent to surgical hand scrub to ensure proper surgical hand preparation. Surgical hand rub is a more practical and quick method, which uses an alcohol-based solution to kill microorganisms on the hands. This method saves time and reduces reliance on water and soap, which may not always be available in an emergency. In addition, the surgical hand rub method can reduce the risk of cross-contamination when using broken or unsanitary handwashing facilities. Therefore, surgical hand rub and surgical hand scrub can be an important solution to maintain hygiene and safety standards in surgical procedures.

This study has at least two limitations. The first limitation of this study is the single-setting study. It was only conducted in one hospital institution. Secondly, the type of surgery involved in this study was limited to clean and clean-contaminated surgery. Therefore, the result of this study could not be generalized into other settings and contexts. Further research with multiple settings and a broader scope of participants and types of surgery is warranted to obtain more generalizable and representative results.

## 5. Conclusions

This study provides clear evidence that surgical hand scrub and hand rub have the same effect in reducing and maintaining the number of colonies on the hands of healthcare workers until the operation is completed. Both methods have similar levels of effectiveness as long as they are applied correctly. Therefore, the decision to choose one of the two hand preparation methods can actually be adjusted to the conditions and availability of facilities in the institution or place of implementation. In addition, the choice of method must also consider the policies that apply in the institution so that the process runs in line with the rules that have been set. In other words, flexibility in choosing a method is possible as long as the standard hand preparation procedures are accurately followed to ensure the ultimate goal is achieved.

## Figures and Tables

**Figure 1 nursrep-15-00242-f001:**
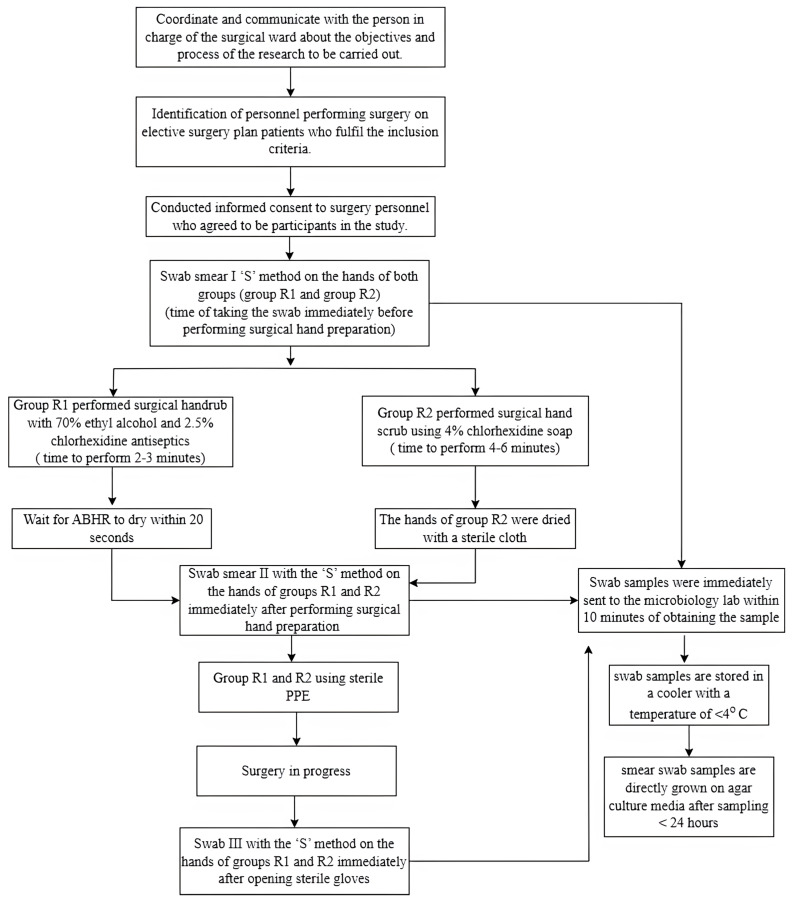
Flow chart of the research process. Note: PPE = Personal Protective Equipment; ABHR = Alcohol Based Hand Rub.

**Figure 2 nursrep-15-00242-f002:**
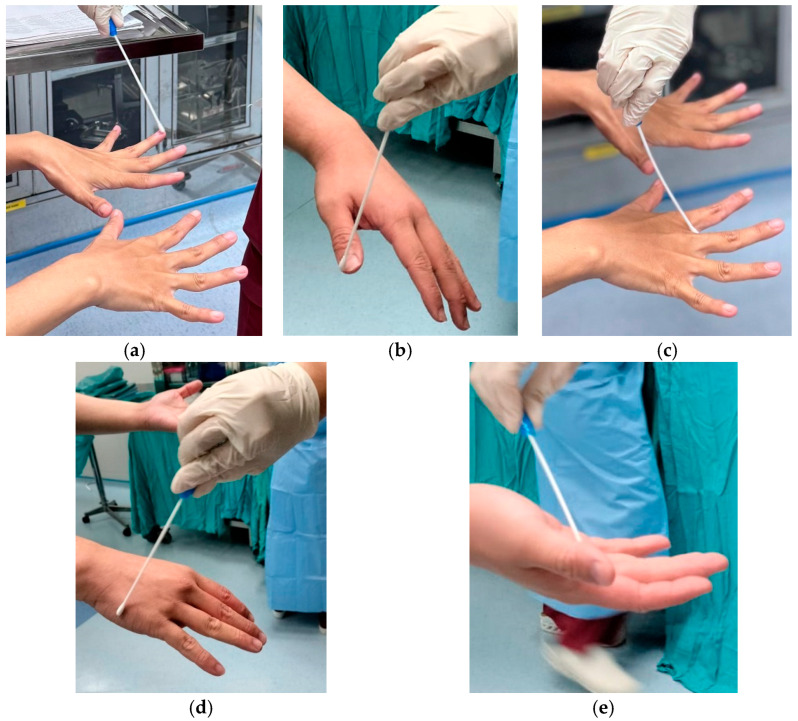
(**a**,**b**) Hand swab in all fingertips area; (**c**) Hand swab in interdigit area; (**d**) Hand swab in dorsal area; (**e**) Hand swab in ventral area.

**Figure 3 nursrep-15-00242-f003:**
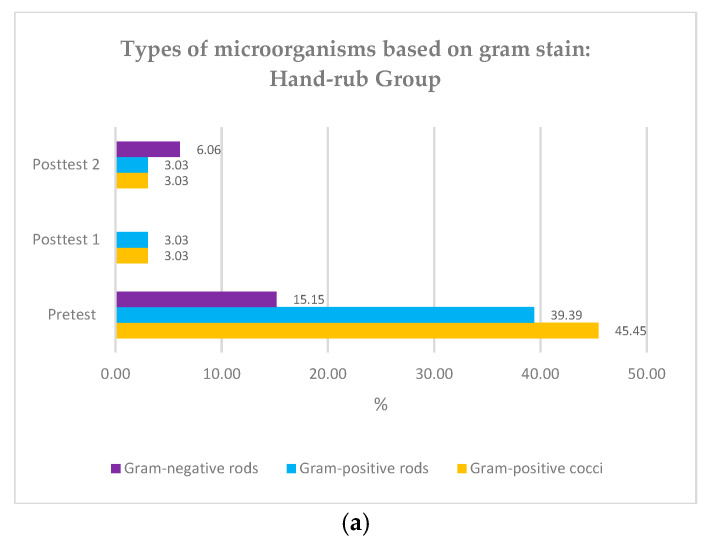
(**a**) Types of microorganisms based on Gram stain: hand rub group; (**b**) Types of microorganisms based on Gram stain: hand scrub group.

**Table 1 nursrep-15-00242-t001:** Participants’ Characteristics.

	Surgical Hand Rub Group		Surgical Hand Scrub Group		*p*
	n	*%*	n	*%*	
Participants Characteristics					
Working Experience					
0–5 year	1	2.9	1	2.8	0.026
6–10 year	0	0	3	4.1	
11–15 year	15	44.1	15	20.3	
16–20 year	16	47.1	11	14.9	
>20 year	2	5.9	6	8.1	
Mean	13.46 ± 5.468		16.17 ± 6.474		
Age					
Mean (year)	40.91 ± 5.711		39.19 ± 8.448		0.996
Minimum (year)	30		27		
Maximum (year)	55		73		
**Procedure Characteristic**					
Duration of Surgery					
≤60 min	15	42.9	15	41.7	0.329
61–120 min	11	31.4	10	27.8	
≥121 min	9	25.7	11	30.6	
Mean (SD)	105 ± 67.47		113.75 ± 63.475		
Profession					
Nurse	18	51.4	33	91.7	0.062
Surgeon	17	48.6	3	8.3	
Type of Surgery					
Clean Surgery	21	60	23	63.9	0.886
Clean Contaminated Surgery	14	40	13	36.1	

**Table 2 nursrep-15-00242-t002:** Distribution of percentage of microorganisms’ types on participants hands.

	Surgical Hand Rub Group (n = 33, %)					Surgical Hand Scrub Group (n = 41, %)					*p* (^b^)
	Gram-positive cocci	Gram-positive rods	Gram-negative rods	Negative (−)	*p* (^a^)	Gram-positive cocci	Gram-positive rods	Gram-negative rods	Negative (−)	*p* (^a^)	
Pre-test	45.45	39.39	15.15	0		63.41	12.2	24.39	0		0.312
Post-test1	3.03	3.03	0	93.94	0	12.2	2.44	4.88	80.49	0	0.104
Post-test2	3.03	3.03	6.06	87.88		19.05	7.14	7.14	6.67		0.051

Note: (^a^) Friedman test; (^b^) Mann–Whitney test; all data are expressed as a percentage (%).

**Table 3 nursrep-15-00242-t003:** Friedman Test and Post Hoc Analysis with Wilcoxon.

	Mean Rank	Mean (CFU/cm^2^)	SD	χ^2^	*p*					
Surgical Hand Scrub Group										
Pre-test	2.89	0.202	0.295	53.397	0.00	(pre-test vs. post-test 1: 0.00; pre-test vs. post-test 2: 0.00; post-test 1 vs. post-test 2: 0.218)				
Post-test 1	1.49	0.007	0.023							
Post-test 2	1.63	0.016	0.043							
**Surgical Hand Rub Group**										
Pre-test	2.81	0.163	0.218	53.109	0.00	(pre-test vs. post-test 1: 0.00; pre-test vs. post-test 2: 0.00; post-test 1 vs. post-test 2: 0.317)				
Post-test 1	1.56	0.001	0.002							
Post-test 2	1.63	0.001	0.004							

**Table 4 nursrep-15-00242-t004:** Comparison of Delta Colony Counts in Both Surgical Hand scrub and Surgical Hand rub Groups: Mann–Whitney Test.

	Surgical Hand Scrub	Surgical Hand Rub	*p*-Value
Delta 1 (pre-test and post-test 1)			0.712
Mean	0.195 (0.295)	0.162 (0.218)	
Std. Deviation			
Delta 2 (pre-test and post-test 2)			0.813
Mean	0.187 (0.284)	0.162 (0.218)	
Std. Deviation			
Delta 3 (post-test 1 and post-test 2)			0.321
Mean	−0.008 (0.046)	0.000 (0.005)	
Std. Deviation			

## Data Availability

The research data from this study can be requested from author via email: claudia23001@mail.unpad.ac.id.

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
