# Peer review of "Which Surgical Hand Preparation Method Is More Effective? A Comparative Study of Hand Rub and Hand Scrub"

_nursrep, 2025, doi:10.3390/nursrep15070242_

Round 1
Reviewer 1 Report
Comments and Suggestions for Authors
I would like to thank the authors for their efforts in addressing an important topic in surgical infection control. The comparison of handrub and handwashing techniques is of great relevance to clinical practice and contributes to the ongoing debate on optimizing surgical hand antisepsis. With this in mind, I would like to offer the following comments and suggestions to help strengthen the manuscript and improve its clarity, methodological rigor, and overall impact:
- The manuscript refers to the study population as including "surgical nurses and instrument nurses." It would be beneficial to clarify the specific roles encompassed by these terms. In particular, the term instrument nurses may not be the most appropriate terminology for referring to scrub nurses. If by surgical nurses the authors are referring to circulating nurses rather than scrub nurses, it is unclear why they were included as participants in a study focused on surgical hand preparation. Further clarification is warranted.
- The introduction appears to have an incomplete review of the existing literature comparing various surgical handrub methods to traditional surgical hand washing (scrubbing). Expanding this section with a broader and more comprehensive review of relevant studies would strengthen the rationale for the research.
- In Section 2.4, "Intervention and Data Collection," the manuscript does not clearly describe the nature of the intervention and control conditions. Specifically, the type and concentration of antiseptic agents used in both the handrub and handwash groups are not specified. This information should be clearly reported in this section and I should not have to look for it elsewhere in the manuscript.
- The flowchart presented in Figure 1 appears to complicate rather than facilitate the understanding of the study design. A revision to improve clarity and visual simplicity is recommended if possible.
- The manuscript states that "the two groups did not show significant differences except in terms of work experience," with the handwash-scrub group having a longer average work experience. This introduces a significant source of bias. Moreover, the study design is not clearly described—specifically, the method of group allocation and the criteria used to assign participants to either the handrub or handwash group are not explained. This lack of transparency undermines the internal validity of the study and should be addressed.
The manuscript contains grammatical and typographical errors, and the overall quality of scientific English is suboptimal. A thorough language revision by a professional editor is advised to enhance readability and academic tone.
Author Response
Response to Reviewer 1's Comments
- Comments 1: In particular, the term instrument nurses may not be the most appropriate terminology for referring to scrub nurses. If by surgical nurses the authors are referring to circulating nurses rather than scrub nurses, it is unclear why they were included as participants in a study focused on surgical hand preparation. Further clarification is warranted.
Response 1: Thank you for pointing this out. We clarify that Instrument Nurses (scrub nurses) manage and handle sterile instruments during surgery, while Assistance Nurses assist the surgeon technically during the operation. Both have direct contact with sterile fields. In contrast, Circulating Nurses do not handle sterile instruments directly but support the surgical environment and coordination. Thus, circulating nurses were not included in our study focused on surgical hand preparation. If the term 'instrument nurse' is uncommon, we have replaced it with 'scrub nurses'.
- Comments 2: The introduction appears to have an incomplete review of the existing literature comparing various surgical handrub methods to traditional surgical hand washing (scrubbing). Expanding this section with a broader and more comprehensive review of relevant studies would strengthen the rationale for the research.
Response 2: Thank you for your valuable feedback. We have expanded and deepened the literature review in the introduction by adding relevant studies comparing surgical handrub methods and traditional surgical scrubbing to strengthen the basis and justification of this research, which can be seen in lines 62-75.
- Comments 3: In Section 2.4, "Intervention and Data Collection," the manuscript does not clearly describe the nature of the intervention and control conditions. Specifically, the type and concentration of antiseptic agents used in both the handrub and handwash groups are not specified. This information should be clearly reported in this section and I should not have to look for it elsewhere in the manuscript.
Response 3: Thank you for pointing this out. We have added the formula of the antiseptic used on line 133-139.
- Comments 4: The flowchart presented in Figure 1 appears to complicate rather than facilitate the understanding of the study design. A revision to improve clarity and visual simplicity is recommended if possible.
Response 4: Thank you for your valuable feedback. We have revised Figure 1 to improve its clarity and visual simplicity. The updated flowchart now offers a clearer and more straightforward representation of the study design to facilitate better understanding.
- Comments 5: The manuscript states that "the two groups did not show significant differences except in terms of work experience," with the handwash-scrub group having a longer average work experience. This introduces a significant source of bias.
Response 5: Thank you for your comment regarding the potential bias due to differences in work experience. It is true that the handwash-scrub group had a longer average work experience, which might influence the results because hand preparation techniques may have evolved over time. However, previous studies have shown that length of service does not significantly affect the implementation of surgical hand preparation. Additionally, other characteristics between the two groups did not differ significantly, suggesting that the bias from work experience is likely minimal. This issue has also been discussed in detail in the discussion section (lines 246-255). We acknowledge this limitation and recommend future studies to control for this factor more thoroughly.
- Comments 6: Moreover, the study design is not clearly described—specifically, the method of group allocation and the criteria used to assign participants to either the handrub or handwash group are not explained. This lack of transparency undermines the internal validity of the study and should be addressed.
Response 6: Thank you for your comment. We would like to clarify that the group allocation was based on the actual opportunity for surgical hand preparation. Participants who performed surgical hand scrub according to protocol were assigned to the surgical hand scrub group, and those who performed surgical handrub according to protocol were assigned to the surgical handrub group. There was no randomization or coercion involved; the allocation was observational and reflected real practice. This procedure has been described in the Methods section to enhance transparency and internal validity. This can be seen in lines 118-123, which discuss the inclusion and exclusion criteria of the samples, as well as lines 124-135, which describe the allocation of samples into the two groups (surgical hand scrub and surgical handrub).
|
Response to Comments on the Quality of English Language |
|
Point 1: The manuscript contains grammatical and typographical errors, and the overall quality of scientific English is suboptimal. A thorough language revision by a professional editor is advised to enhance readability and academic tone. |
|
Response 1: We would like to express our gratitude for the valuable feedback provided. We have thoroughly revised the manuscript to correct the grammatical and typographical errors identified. We believe these revisions have significantly improved the overall quality of the manuscript.
|

Reviewer 2 Report
Comments and Suggestions for Authors
- The title has a punctuation error. The question mark should not be followed by a colon. This is because a question mark signals the end of a sentence, and no other punctuation, including a colon, should follow it.
- “Handwash-scrub with antimicrobial soap” is not commonly used in published hand hygiene guidelines. Replace it with “antimicrobial soap for surgical hand scrub.” Refer to https://www.ncbi.nlm.nih.gov/books/NBK144046 for more details. Replace all other instances of “handwash-scrub with antimicrobial soap” in the text accordingly.
- Similarly, the “surgical handwash-scrub group” can be renamed to “surgical hand scrub group.”
- Line 18: “Germ” is a layman term – it’s usage should be avoided in scholarly articles. There are also other occurrences of “germ” in the manuscript that should be corrected.
- Line 20: what are these so called “two groups?”
- Lines 23-25: what is the unit for these colony counts?
- Lines 23-25: the wordings used to present the study’s findings need to be rephrased. Presently, it appears too crude. A suggested rephrase is “The surgical hand scrub group had average pretest, posttest 1 and posttest 2 colony counts of 0.202, 0.007 and 0.016, respectively….” The same format can be applied to the hand rub group as well.
- Lines 30-32: It is unusual to have a “recommendation” subheading in the abstract. It can be deleted.
- The formulation of the hand rub and hand scrub were not mentioned in the text under materials and methods.
- The method of performing the total plate count was not provided. This information is essential for reproducibility. Also, is the colony count unit CFU/ml or CFU/g?
- Line 81: the sentence “The population in this study was surgery…” sounds incorrect and needs to be rephrased.
- Figure 1 has no legend even though acronyms (PPE, ABHR) were used.
- Table 1: the unit for age was not specified.
- Table 1: why was “f” (frequency) used instead of “n” (number)?
- Table 2: inconsistent portrayal of the decimal point. For one number (i.e. 3.03) a period was used, and for the rest, a comma was used.
- Table 2: the “%” heading is misplaced. The correct heading here should be “time of sampling” or something along this line. In the footnote, it should be stated that all data are expressed as percentage unless stated otherwise.
- Table 2: what is the meaning of an “o” with a slash?
- Table 2: what is “uji" Mann Whitney”
- Is Figure 2 really necessary? It contains the same data as in Table 2.
- Table 3: what is the unit for the colony counts?
This manuscript contains numerous grammatical errors. Proofreading by an English language expert should have been done before submission.
Author Response
Response to Reviewer 2's Comments
Comments 1: The title has a punctuation error. The question mark should not be followed by a colon. This is because a question mark signals the end of a sentence, and no other punctuation, including a colon, should follow it.
Response 1: Thank you for pointing this out. We have fixed this part.
Comments 2: “Handwash-scrub with antimicrobial soap” is not commonly used in published hand hygiene guidelines. Replace it with “antimicrobial soap for surgical hand scrub.” Refer to https://www.ncbi.nlm.nih.gov/books/NBK144046 for more details. Replace all other instances of “handwash-scrub with antimicrobial soap” in the text accordingly.
Response 2: Thank you for pointing this out. We have fixed this part.
Comments 3: Similarly, the “surgical handwash-scrub group” can be renamed to “surgical hand scrub group.”
Response 3: Thank you for pointing this out. We have fixed this part.
Comments 4: Line 18: “Germ” is a layman term – it’s usage should be avoided in scholarly articles. There are also other occurrences of “germ” in the manuscript that should be corrected.
Response 4: Thank you for pointing this out. we have changed the term germ to bacteria.
Comments 5: Line 20: what are these so called “two groups?”
Response 5: Thank you for pointing this out. Two groups refer to the surgical hand scrub group and the surgical handrub group. We have added the caption.
Comments 6: Lines 23-25: what is the unit for these colony counts?
Response 6: Thank you for pointing this out. The colony counts in this study is colont forming units (CFU) and we have added the unit.
Comments 7: Lines 23-25: the wordings used to present the study’s findings need to be rephrased. Presently, it appears too crude. A suggested rephrase is “The surgical hand scrub group had average pretest, posttest 1 and posttest 2 colony counts of 0.202, 0.007 and 0.016, respectively….” The same format can be applied to the hand rub group as well.
Response 7: Thank you for pointing this out. We have revised the sentence and agree with the suggestion made.
Comments 8: Lines 30-32: It is unusual to have a “recommendation” subheading in the abstract. It can be deleted.
Response 8: Thank you for pointing this out. We have corrected it.
Comments 9: The formulation of the hand rub and hand scrub were not mentioned in the text under materials and methods.
Response 9: Thank you for pointing this out. The formulations of handrub and hand scrub have been described in figure 1 and in the discussion section. But we have added the formulation in the abstract.
Comments 10: The method of performing the total plate count was not provided. This information is essential for reproducibility. Also, is the colony count unit CFU/ml or CFU/g?
Response 10: Thank you for pointing this out. In this study, the microbiology analyst directly inoculated samples obtained using Amies swabs onto Nutrient agar media. After incubation under specified conditions, the colonies that developed were enumerated by assessing the surface area occupied by the colony growth on the agar medium. The colony counts are reported as Colony Forming Units per square centimeter (CFU/cm²). This method was chosen to provide representative results consistent with the sample type and collection technique used, thereby ensuring the reproducibility of the study. For completeness, we have added a method to perform the total plate count calculation on lines 148-159.
Comments 11: Line 81: the sentence “The population in this study was surgery…” sounds incorrect and needs to be rephrased.
Response 11: Thank you for pointing this out. We have revised the sentence originally written as “The population in this study was surgery…” to improve clarity and correctness. The sentence is now rephrased to: "The population included in this study..."
Comments 12: Figure 1 has no legend even though acronyms (PPE, ABHR) were used.
Response 12: Thank you for pointing this out. We have added a legend just below figure 1
Comments 13: Table 1: the unit for age was not specified.
Response 13: Thank you for pointing this out. We have added units to the age section
Comments 14: Table 1: why was “f” (frequency) used instead of “n” (number)?
Response 14: Thank you for pointing this out. In our manuscript, we used ‘f’ to denote the frequency (the count of occurrences) of each category or group within the sample population. This is a common notation in descriptive statistics to emphasise the counted occurrences. However, we understand that ‘n’ is also widely accepted and used to represent the number of subjects or items in a group. If preferred, we are willing to revise the notation to ‘n’ for consistency and clarity throughout the manuscript.
Comments 15: Table 2: inconsistent portrayal of the decimal point. For one number (i.e. 3.03) a period was used, and for the rest, a comma was used.
Response 15: We acknowledge the inconsistent depiction of the decimal point. We used period (dots) as the decimal separator throughout the document and it has been fixed
Comments 16: Table 2: the “%” heading is misplaced. The correct heading here should be “time of sampling” or something along this line. In the footnote, it should be stated that all data are expressed as percentage unless stated otherwise.
Response 16: Thank you for pointing this out. we have put “%” on the sample line and completed a small note below the table stating that all data is expressed in percentage.
Comments 17: Table 2: what is the meaning of an “o” with a slash?
Response 17: Thank you for pointing this out. ‘ø’ in the table indicates “no data” or in the context of our study indicates no microorganisms were found. If ‘ø’ is not commonly used, we have changed it to negative.
Comments 18: Table 2: what is “uji" Mann Whitney”
Response 18: Thank you for pointing this out. Uji means test, which is our negligence, so this word is not translated. We have replaced the word with English.
Comments 19: Is Figure 2 really necessary? It contains the same data as in Table 2.
Response 19: Thank you for pointing this out. Figure 2 and table 2 do have the same data. However, we would like to show the graphical form to the reader for easier understanding.
Comments 20: what is the unit for the colony counts?
Response 20: Thank you for pointing this out. The unit of colony count is CFU/cm2. We have corrected the table by adding units.
|
Response to Comments on the Quality of English Language |
|
Point 1: This manuscript contains numerous grammatical errors. Proofreading by an English language expert should have been done before submission. |
|
Response 1: We would like to express our gratitude for the valuable feedback provided. We have thoroughly revised the manuscript to correct the grammatical errors identified. We believe these revisions have significantly improved the overall quality of the manuscript.
|

Reviewer 3 Report
Comments and Suggestions for Authors
Considering the increased burden of antimicrobial resistance globally, the topic is relevant to healthcare providers, policymakers, and the general public. The title reflects the content of the manuscript. The abstract correctly summarises the content of the manuscript. The introduction orients the readers to the topic, identifies a research gap, and states the aim of the study. The material and methods section requires improvement. The results are clearly presented with the help of tables and figures. The authors compare their findings with previous literature in the discussion, while explaining their results with the help of the literature. The authors also acknowledge the limitations of their study. The conclusion is based on the study’s findings.
MAJOR REVISIONS
Title
- The title should read: Which surgical hand preparation method is more effective?: A comparative study of handrub and handwash.
Keywords
- Add effectiveness to the keywords.
Introduction
- In lines 44-45, the authors state, ‘Study from Nthumba et al showed the average incidence of damage gloves was 18% during surgery and 35% after….’ Change ‘damage’ to ‘damaged’
- In lines 54-55, the authors state, ‘Alcohol-based antiseptic products are much more effective than antiseptic soap products (11).’ This statement contradicts some of the information in the introduction. The authors should rather state that it was a finding of a study.
- The authors must provide more information on the practicality of using either method.
Materials and Methods
- In line 81, the authors state, ‘The population in this study was surgery…’ Surgery cannot be the population. The population in this study should be surgical personnel who provided the samples. This should be corrected.
- In lines 86-88, the authors state, ‘From previous research conducted by Villa et al (2020) entitled “Comparing Rubbing and Scrubbing Surgical Hand Antiseptic with Propan-1-Ol 60%.’ These sentences are incomplete, so they should be rephrased.
- The authors must clearly state the inclusion and exclusion criteria for taking part in the study.
- In lines 103-104, the authors state, ‘Those who willing to participate in this study completed a consent form.’ Add ‘were’ after ‘who.’
Conclusion
- The authors must add a sentence on the recommendations from the findings in the conclusion.
References
- Please ensure that all references in the reference list adhere to journal guidelines.
Author Response
Response to Reviewer 3's Comments
Title
Comments 1: The title should read: Which surgical hand preparation method is more effective?: A comparative study of handrub and handwash.
Response 1: Thank you for pointing this out. We have fixed this part.
Keywords
Comments 2: Add effectiveness to the keywords.
Response 2: Thank you for pointing this out. We have added keywords.
Introduction
Comments 3: In lines 44-45, the authors state, ‘Study from Nthumba et al showed the average incidence of damage gloves was 18% during surgery and 35% after….’ Change ‘damage’ to ‘damaged’
Response 3: Thank you for pointing this out. We have fixed this part.
Comments 4: In lines 54-55, the authors state, ‘Alcohol-based antiseptic products are much more effective than antiseptic soap products (11).’ This statement contradicts some of the information in the introduction. The authors should rather state that it was a finding of a study.
Response 4: Thank you for pointing this out. We have added supporting previous research on lines 58-59.
Comments 5: The authors must provide more information on the practicality of using either method.
Response 5: Thank you for pointing this out. We have added information on the cost and time to implement both methods, which can be seen on lines 62-75.
Materials and Methods
Comments 6: In line 81, the authors state, ‘The population in this study was surgery…’ Surgery cannot be the population. The population in this study should be surgical personnel who provided the samples. This should be corrected.
Response 6: Thank you for pointing this out. we have corrected the population in this study which can be seen on line 100-106
Comments 7: In lines 86-88, the authors state, ‘From previous research conducted by Villa et al (2020) entitled “Comparing Rubbing and Scrubbing Surgical Hand Antiseptic with Propan-1-Ol 60%.’ These sentences are incomplete, so they should be rephrased.
Response 7: Thank you for pointing this out. We have corrected it. There should be no full stop punctuation in the middle of the sentence, causing the research title to appear to be 2 sentences.
Comments 8: The authors must clearly state the inclusion and exclusion criteria for taking part in the study.
Response 8: Thank you for pointing this out. We have added inclusion and exclusion criteria for participants taking part in this study on line 118-125.
Comments 9: In lines 103-104, the authors state, ‘Those who willing to participate in this study completed a consent form.’ Add ‘were’ after ‘who.’
Response 9: Thank you for pointing this out. We have corrected it.
Conclusion
Comments 10: The authors must add a sentence on the recommendations from the findings in the conclusion.
Response 10: Thank you for pointing this out. We have added recommendations from the findings of this study. The recommendations in the conclusion section can be found on lines 337-342.
References
Comments 11: Please ensure that all references in the reference list adhere to journal guidelines.
Response 11: Thank you for your attention. The references in this article use the APA Reference Style, which complies with the guidelines of the journal.

Round 2
Reviewer 2 Report
Comments and Suggestions for Authors
In the abstract, the unit for colony counts is still missing.
Use "n" instead of "f" to indicate the number of study subjects in tables.
Apart from these, I have no other comments.
Comments on the Quality of English LanguageAlthough the authors claim the paper had been proofread by an English language expert before its resubmission, some very minor grammatical errors still persist.
Author Response
Round 2: Response to Reviewer 2's Comments
Comments 1: In the abstract, the unit for colony counts is still missing.
Response 2: Thank you for pointing this out. We have revised the manuscript and added units in the abstract section on line 27-28.
Comments 2: Use "n" instead of "f" to indicate the number of study subjects in tables.
Response 2: Thank you very much for your valuable feedback and suggestions. We have corrected it by replacing 'f' with 'n' in the numbers presented in the table
|
Response to Comments on the Quality of English Language |
|
Point 1: Although the authors claim the paper had been proofread by an English language expert before its resubmission, some very minor grammatical errors still persist.
|
|
Response 1: We would like to express our gratitude for the valuable feedback provided. We have carefully reviewed and corrected the grammar throughout the manuscript.
|
Reviewer 3 Report
Comments and Suggestions for Authors
The authors have satisfactorily responded to all my comments.
Author Response
Round 2: Response to Reviewer 3's Comments
Comments 1: The authors have satisfactorily responded to all my comments.
Response 1: We sincerely thank the reviewer for the positive evaluation and are pleased to hear that our responses have satisfactorily addressed all comments